# Long-Term Functional Outcomes and Quality of Life Following Carpal Tunnel Release Surgery

**DOI:** 10.3390/ijerph21091203

**Published:** 2024-09-11

**Authors:** Yuval Krieger, Mai Ofri, Gali Sagi, Hila Moshe-Noach, Pnina Raiden, Yaron Shoham, Amiram Sagi, Anat Reiner Benaim, Eldad Silberstein

**Affiliations:** 1Hand Unit, Plastic and Reconstructive Surgery, Soroka Medical Center, Faculty of Health Sciences, Ben-Gurion University of the Negev, Beer Sheva P.O. Box 151, Israel; yshoham@bgu.ac.il (Y.S.); amiramsagi@gmail.com (A.S.); eldads@bgu.ac.il (E.S.); 2Soroka Clinical Research Center, Soroka University Medical Center, Faculty of Health Sciences, Ben-Gurion University of the Negev, Beer Sheva P.O. Box 151, Israel; maiof@post.bgu.ac.il; 3Department of Public Health Epidemiology, Biostatistics and Community Health, Faculty of Health Sciences, Ben-Gurion University of the Negev, Beer Sheva P.O. Box 151, Israel; gali.gaash@gmail.com (G.S.); hillamn73@gmail.com (H.M.-N.); pnina.gas86@gmail.com (P.R.); reinera@bgu.ac.il (A.R.B.)

**Keywords:** carpal tunnel syndrome, carpal tunnel release surgery, long-term outcomes, BCTQ, quality of life, hand function, functional limitations

## Abstract

This study investigates the long-term efficacy of carpal tunnel release surgery (CTR) on patient outcomes. We aimed to assess symptom severity, functional limitations, and quality of life in a large cohort (n = 186) at least five years post-surgery via a retrospective cross-sectional design that evaluated participants using the validated Boston Carpal Tunnel Questionnaire (BCTQ) over a phone interview. The BCTQ measures symptom severity, functional limitations, and quality of life specific to carpal tunnel syndrome. Seventy-three percent (73.1%) of patients reported complete resolution of symptoms and functional limitations (BCTQ = 1) with an average follow-up of 11 years. No statistically significant decline in BCTQ scores was observed over time. Univariate analysis revealed a significant association between poorer outcomes and older age at surgery and current unemployment. Carpal tunnel release surgery demonstrates long-term effectiveness in relieving symptoms and improving function in most patients. These findings contribute to the understanding of CTR’s impact on patient well-being beyond short-term outcomes.

## 1. Introduction

Carpal tunnel syndrome (CTS) is a prevalent condition characterized by median nerve compression within the carpal tunnel, a narrow passage at the wrist. Symptoms typically include pain, paresthesia, and numbness in the thumb, index, and middle fingers. The condition affects an estimated 3% to 8% of the general population, with a higher incidence among women and individuals aged 40–60 years.

The etiology of CTS is multifactorial. While increased carpal tunnel pressure is a key factor, the precise cause remains elusive. Both anatomic compression and inflammation are implicated. Risk factors include genetic predisposition, diabetes mellitus, arthritis, obesity, hypothyroidism, pregnancy, wrist trauma, aromatase inhibitor use, and environmental exposures [1,2].

Diagnosis is established through physical examination and nerve conduction studies. Treatment modalities encompass conservative and surgical approaches. Non-surgical interventions include rest, splinting, patient education, neurodynamic therapy, anti-inflammatory medications, heat therapy, massage, ultrasound, acupuncture, and corticosteroid injections. Carpal tunnel release surgery, either open or endoscopic, aims to alleviate median nerve compression [3].

While surgery often provides short-term symptom relief and functional improvement, long-term outcomes beyond two years are less well defined. Moreover, the influence of age, gender, and comorbidities on long-term surgical results remains inconsistent across studies [4]. This investigation aims to comprehensively evaluate the long-term efficacy of carpal tunnel release surgery in a substantial patient cohort, identifying potential predictors of surgical success.

## 2. Materials and Methods

### 2.1. Study Design and Participants

A retrospective cross-sectional study was conducted to assess the long-term outcomes of carpal tunnel release (CTR) surgery. Patients who underwent CTR at least five years prior to enrollment were included. The exclusion criteria encompassed pregnant women, individuals under 18 years, and those with impaired judgment, communication difficulties, or dementia. Eligible participants were contacted by telephone up to three times. Informed consent was obtained prior to a telephone interview.

### 2.2. Outcome Measures

Patient-reported outcomes were assessed using the Boston Carpal Tunnel Questionnaire (BCTQ), a validated instrument measuring symptom severity (SSS) and functional status (FSS) [5,6,7]. The SSS comprises 11 items evaluating pain, paresthesia, numbness, weakness, nocturnal symptoms, and grasping difficulties. The FSS consists of eight items assessing functional limitations in daily activities. Scores for both scales range from 1 (no symptoms/difficulties) to 5 (severe symptoms/unable to perform activity), with higher scores indicating worse outcomes.

### 2.3. Statistical Analysis

Descriptive statistics were employed to summarize sociodemographic and clinical characteristics and BCTQ scores. Means ± standard deviations, medians, minimums, and maximums were calculated for continuous variables, while frequencies and percentages were reported for categorical variables. Univariate comparisons were conducted using the Chi-square test for categorical data and the Wilcoxon rank-sum test for continuous data. A generalized linear model incorporating variables with *p*-values < 0.05 from univariate analysis and informed by clinical expertise was employed for multivariate analysis. To visualize the temporal trend of mean BCTQ scores post-surgery, locally estimated scatterplot smoothing (LOESS) with 95% confidence intervals was utilized. Statistical analyses were performed using R version 4.3.1.

## 3. Results

During the study period, 186 patients consented to participate and completed the interview, representing a 44.9% response rate from 414 eligible patients. The remaining 229 individuals were excluded due to inability to contact (121), refusal to participate (40), communication difficulties (45), or incomplete interviews (22).

This study aimed to evaluate patient satisfaction following carpal tunnel release surgery and compare outcomes between subgroups. A post hoc sample size calculation was performed to ensure adequate statistical power for these comparisons. Assuming a conservative 50% satisfaction rate and a desired power of 80% at a significance level of 0.05, a sample size of 186 patients was determined to be sufficient for group comparisons.

The average age at surgery was 53.9 years, with 29% males. The median follow-up period was 11 years. Most participants were married (71.2%) and employed (66.5%), with 53% being aged 67 or older and 19% previously unemployed.

Table 1 compares patient characteristics based on symptom status (BCTQ = 1 vs. BCTQ > 1). Patients with symptoms were older at surgery (55.0 ± 10.5 vs. 51.4 ± 11.2 years, *p* = 0.009) and less likely to be employed (22% vs. 83%, *p* < 0.001) or retired (31% vs. 62%, *p* < 0.001). Unemployment before retirement age was higher in the symptomatic group (49% vs. 8.1%, *p* < 0.001).

Among the 186 participants, 73.1% (n = 136) reported no symptoms (BCTQ = 1). The remaining patients reported BCTQ scores as follows: 1.01–2.0 (9.1%, n = 17), 2.01–3.0 (8.1%, n = 15), 3.01–4.0 (4.8%, n = 9), and 4.01–5.0 (4.8%, n = 9) (Table 2).

BCTQ scores were not associated with gender (OR 1.54, 95% CI 0.71–3.53, *p* = 0.285) or diabetes (OR 1.42, 95% CI 0.64–3.32, *p* = 0.398) after adjusting for age at surgery (Table 3). However, unemployment was associated with higher symptom scores.

Figure 1 illustrates the mean BCTQ score over time since surgery, demonstrating a temporal trend. Figure 2 presents the distribution of symptom status by time since surgery.

## 4. Discussion

Carpal tunnel release (CTR) is a surgical intervention aimed at alleviating carpal tunnel syndrome (CTS) symptoms by dividing the transverse carpal ligament, thereby decompressing the median nerve. While generally safe, CTR carries potential risks including bleeding, infection, nerve or tendon injury, scar-related issues, and persistent pain. Factors influencing short-term outcomes encompass symptom severity, surgical expertise, and postoperative care, whereas long-term success hinges on nerve recovery and sustained function.

Our study demonstrates the long-term efficacy of CTR, with high patient satisfaction rates up to 15 years post surgery. Most participants reported significant symptom improvement, reduced pain, and enhanced quality of life, as evidenced by the high proportion (73.1%) achieving a BCTQ score of 1. A negative correlation between time since surgery and BCTQ scores was observed (Figure 1).

Consistent with our findings, previous research has reported favorable long-term outcomes following CTR. Nancollas et al. found good or excellent outcomes in 87% of patients at an average of 5.5 years post surgery, although some residual symptoms persisted [8]. Shai et al. reviewed seven controlled studies and concluded that surgical treatment has a superior benefit, in terms of symptoms and function, at six and twelve months [9]. Tang et al. demonstrated positive outcomes and high satisfaction rates in patients with bilateral severe CTS at a mean follow-up of nine years [10], while other reviews confirmed significant improvements in clinical and subjective outcomes, independent of patient characteristics [11,12]. De Kleermaeker et al. also reported favorable outcomes in 81.6% of patients at a median follow-up of nine years, emphasizing the importance of early treatment success [13]. Huistestede et al. presented an evidence-based overview of the effectiveness of surgical and post-surgical interventions for CTS and stated that for the long term, surgery was more effective than noninvasive treatment [14], and Alimohammadi et al. identified predictors of better outcomes, including younger age, shorter symptom duration, and higher preoperative function [15].

Limitations: The retrospective cross-sectional design precludes causal inference and limits the assessment of temporal relationships. The reliance on self-reported outcomes introduces potential bias, and the absence of a control group hinders direct comparison with the natural history of CTS. Additionally, objective nerve function measures were lacking. Furthermore, the present study is limited by the absence of data from patients who did not respond to the survey, restricting our ability to characterize this non-respondent group. To further elucidate the long-term outcomes and identify optimal patient candidates for carpal tunnel release surgery, a prospective cohort study with extended follow-up is warranted.

Despite these limitations, our findings provide compelling evidence for the long-term benefits of CTR in alleviating CTS symptoms and improving patient-reported outcomes.

## 5. Conclusions

Carpal tunnel release (CTR) is an established surgical treatment for carpal tunnel syndrome, providing significant relief from symptoms and improved function in most patients. Our study findings underscore the long-term efficacy of CTR, demonstrating sustained benefits in symptom reduction, functional improvement, and overall quality of life for up to 15 years post surgery.

## Figures and Tables

**Figure 1 ijerph-21-01203-f001:**
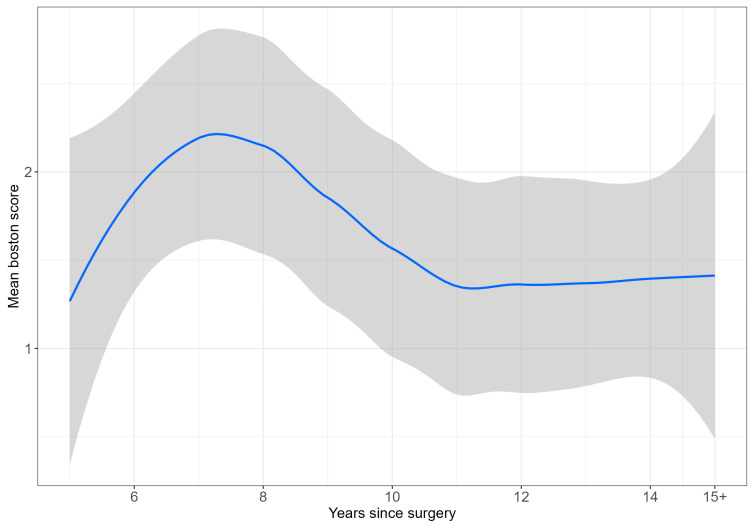
Mean BCTQ score, by years since surgery (The blue liner represents the mean BCTQ score value while the Gray zone represents the 95% confidence interval).

**Figure 2 ijerph-21-01203-f002:**
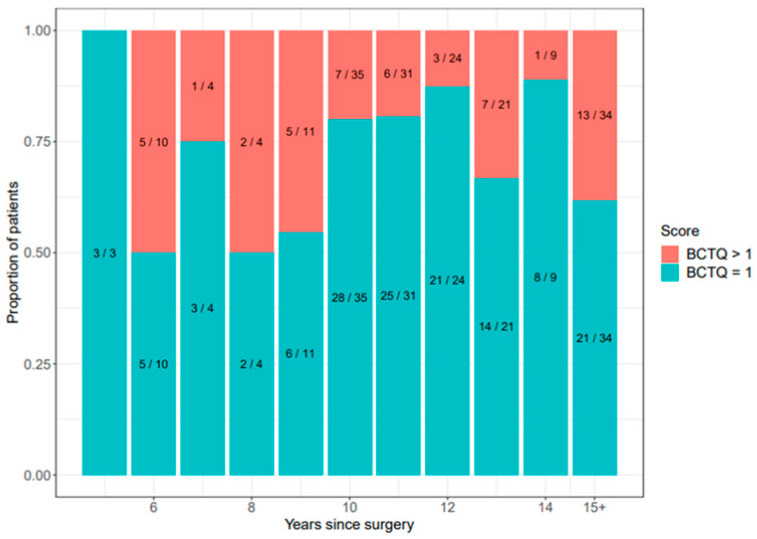
Proportion of patients with BCTQ score = 1 and BCTQ score > 1, by years since surgery.

**Table 1 ijerph-21-01203-t001:** Background characteristics table, stratified by BCTQ status.

	Overall, N = 186 ^1^	BCTQ = 1, N = 136 ^1^	BCTQ > 1, N = 50 ^1^	*p*-Value ^2^
Female	132/186 (71%)	94/136 (69%)	38/50 (76%)	0.359
Age at surgery				0.009
Mean, (No. obs.)	54, (186)	55, (136)	51, (50)	
Median	55	57	51	
Range	24, 88	31, 88	24, 81	
Years since surgery				0.837
Mean, (No. obs.)	12.1, (186)	11.8, (136)	12.6, (50)	
Median	11.0	11.0	11.0	
Range	5.0, 27.0	5.0, 27.0	6.0, 24.0	
Marital status				0.069
Married	116/163 (71%)	88/117 (75%)	28/46 (61%)	
Single	47/163 (29%)	29/117 (25%)	18/46 (39%)	
Employed	123/185 (66%)	112/135 (83%)	11/50 (22%)	<0.001
Retirement age at survey	87/165 (53%)	72/116 (62%)	15/49 (31%)	<0.001
Unemployed and before retirement age	35/184 (19%)	11/135 (8.1%)	24/49 (49%)	<0.001
Diabetes	49/186 (26%)	38/136 (28%)	11/50 (22%)	0.415

^1^ n/N (%); ^2^ Pearson’s Chi-squared test; Wilcoxon rank-sum test.

**Table 2 ijerph-21-01203-t002:** Post surgical BCTQ score values.

	SSS ^1,2^	FSS ^1,2^	BCTQ ^1,2^
Score	1.00 (1.00, 1.00)	1.00 (1.00, 1.00)	1.00 (1.00, 1.20)
Score—categorical			
1	140 (75%)	140 (75%)	136 (73%)
≤2	16 (8.6%)	14 (7.5%)	17 (9.1%)
≤3	16 (8.6%)	12 (6.5%)	15 (8.1%)
≤4	7 (3.8%)	9 (4.8%)	9 (4.8%)
≤5	7 (3.8%)	11 (5.9%)	9 (4.8%)

^1^ Median (IQR); n (%); ^2^ symptom severity scale, functional status scale, Boston Carpal Tunnel Syndrome Questionnaire.

**Table 3 ijerph-21-01203-t003:** Logistic regression to estimate association of background variables with a BCTQ score of 1.

Characteristic	OR ^1^	95% CI ^1^	*p*-Value
Age at surgery	1.03	1.00, 1.07	0.082
Diabetes	1.42	0.64, 3.32	0.398
Gender	0.65	0.28, 1.40	0.285

^1^ OR = odds ratio, CI = confidence interval.

## Data Availability

Data supporting reported results will be provided by the corresponding author upon request.

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
