# Peer review of "Long-Term Functional Outcomes and Quality of Life Following Carpal Tunnel Release Surgery"

_ijerph, 2024, doi:10.3390/ijerph21091203_

Round 1
Reviewer 1 Report
Comments and Suggestions for Authors
I am not qualified to discuss the clinical part, so the comments below are about the data/analysis presentation.
Row 115 "BCTQ score was not associated with patients' gender or history of diabetes...": the authors did not provide the results of the model (coefficients? p values?). Add a table with the results.
Figure 1 is useless, these information can be found in Table 2 (that can be inserted in table 1...).
Figures 2 and 3 make categories of years since surgery, but some categories are very small (3 subjects), so the analysis will not be consistent. The years are a continuous variable, please analyze it as that.
The groups of BCTQ (=1 and >1) are very different in numerosity (1:3), please discuss this fact.
There is no sample size/power calculation.
Typing "carpal tunnel AND surgery AND function" in Pubmed there are more than 2600 articles... I think a bibliography of 13 papers (7 older than 5 years) is insufficient for this topic.
There are a lot of typos, please check them.
Author Response
Comment 1: Row 115 "BCTQ score was not associated with patients' gender or history of diabetes...": the authors did not provide the results of the model (coefficients? p values?). Add a table with the results.
Response 1: We appreciate the reviewer's keen attention to detail. We apologize for the oversight in not explicitly presenting the results of the model assessing the association between BCTQ score, gender, and diabetes.
As requested, we have added a new table (Table 2) to the manuscript, presenting the full model results, including coefficients and p-values. The table clearly demonstrates that neither gender nor diabetes history was significantly associated with BCTQ score after adjusting for age at surgery.
Comment 2: Figure 1 is useless, this information can be found in Table 2 (that can be inserted in table 1...).
Response 2: We appreciate the reviewer's suggestion to improve the manuscript's clarity. We agree that Figure 1 was redundant and has been removed. The information previously presented in the figure is now incorporated into the text (lines 130-134) and detailed in Table 3. We believe that maintaining Table 3 as a separate table effectively highlights the primary outcome of the study, distinct from the general patient characteristics presented in Table 1.
Comment 3: Figures 2 and 3 make categories of years since surgery, but some categories are very small (3 subjects), so the analysis will not be consistent. The years are a continuous variable, please analyze it as that.
Response 3: We thank the reviewer for their insightful comment. We agree that treating years since surgery as a continuous variable would provide a more robust analysis. As suggested, we have removed Figure 2 and replaced it with a smoothed plot using LOESS (now Figure 1) to visualize the relationship between BCTQ score and time since surgery. This approach allows for a more nuanced exploration of the data. Figure 3 has been renumbered to Figure 2.
Comment 4: The groups of BCTQ (=1 and >1) are very different in numerosity (1:3), please discuss this fact.
Response 4: We appreciate the reviewer's comment regarding the imbalance in sample size between the BCTQ groups. As noted in lines 165-170, the high proportion of patients with a BCTQ score of 1 (73%) is indeed indicative of favorable long-term outcomes following carpal tunnel release surgery. While this imbalance might limit the power of certain analyses, it also underscores the effectiveness of the procedure in a substantial portion of the patient population. We will consider further exploring potential confounders or effect modifiers related to this imbalance in future research.
Comment 5: There is no sample size/power calculation.
Response 5: We appreciate the reviewer's attention to detail regarding sample size calculations. A sample size calculation was conducted to ensure adequate power for the study. We have included this information in the revised manuscript lines 149-151.
Comment 6: Typing "carpal tunnel AND surgery AND function" in Pubmed there are more than 2600 articles... I think a bibliography of 13 papers (7 older than 5 years) is insufficient for this topic.
Response 6: We appreciate the reviewer's suggestion to expand the reference list. While the topic of carpal tunnel surgery is indeed extensive, the focus of our study is on long-term outcomes. As such, we have carefully selected references that directly address this specific aspect of carpal tunnel syndrome. In response to the reviewer's comment, we have added two additional studies (references 9 and 14) to the reference list and added reference to their conclusions in the discussion paragraph, (lines 372-386 of the attached revised manuscript) which further support our findings regarding the long-term effects of carpal tunnel release surgery.
Comment 7: There are a lot of typos, please check them.
Response 7: Thank you for bringing the typographical errors to our attention. We have carefully reviewed the manuscript and made extensive corrections. Over 50 typographical and grammatical errors were rectified to improve the overall readability and clarity of the text.

Reviewer 2 Report
Comments and Suggestions for Authors
Dear authors,
thank you for this interesting study on the long-term results of carpal tunnel release.
Introduction:
Line 28-36: There are several repetitions (innervated area of the median nerve; symptoms of carpal tunnel). Please shorten.
Line 36: There is a word missing in the sentence concerning the incidence.
Results:
Line 106: The 33.5 % of patients unemployed were because lacking a job or maybe some retirements?
There might be a bias since older age is correlated with a worse outcome and is also correlated with being unemployed because of retirement.
Line 118: The sum of your reported patients in these lines is 48 instead of 50 in your figure and table.
Table 2: What is the meaning of the first line?
Figure 1: Redundant to Table 2.
Discussion:
Line 137 - 150: Partly redundant to the introduction, please shorten.
Line 151: ... up to 15 years ...
Line 153: Despite it is well know that CTR is a safe technique with low complication rate you cannot state that from your study since you did not analysis the complication rate.
Comments on the Quality of English Language
There are several misspellings, mistakes in upper and lower cases etc. Please revise the manuscript thorougly, preferabaly by a native speaker.
Author Response
Comment 1: Line 28-36: There are several repetitions (innervated area of the median nerve; symptoms of carpal tunnel). Please shorten.
Response 1: We appreciate the reviewer's careful reading of the manuscript. We agree that there was redundancy in lines 28-36 regarding the innervated area and symptoms of carpal tunnel syndrome. We have carefully revised the text to eliminate these repetitions and improve clarity.
Comment 2: Line 36: There is a word missing in the sentence concerning the incidence.
Response 2: Thank you for your careful review of the manuscript. We have re-written the sentence that now reads: " The condition affects an estimated 3% to 8% of the general population, with a higher incidence among women and individuals aged 40-60 years."
Comment 3: Line 106: The 33.5 % of patients unemployed were because lacking a job or maybe some retirements?
There might be a bias since older age is correlated with a worse outcome and is also correlated with being unemployed because of retirement.
Response 3: We appreciate the reviewer's insightful comment regarding the potential bias associated with the high unemployment rate among study participants. To address this concern, we have included additional data on retirement status in lines 215-217 and 225. Specifically, we report that 53% of participants were aged 67 or older and 19% were unemployed before reaching retirement age. While it is possible that retirement may contribute to the observed association between employment status and outcome, our analysis controlled for age at surgery as a potential confounder. We acknowledge the limitations of our study design and plan to explore this issue further in future research.
Comment 4: Line 118: The sum of your reported patients in these lines is 48 instead of 50 in your figure and table.
Response 4: Thank you, corrected the typing error from 13 to 15 patients in the score group between 2 and 3.
Comment 5: Table 2: What is the meaning of the first line?
Response 5: The first line has acronyms of the two parts of the Boston Carpal Tunnel Syndrome Questionnaire. SSS = symptom severity scale, FSS = functional status scale. We added this in the footnote.
Comment 6: Figure 1: Redundant to Table 2.
Response 6: Thank you for this insight. The Figure was omitted.
Comment 7: Line 137 - 150: Partly redundant to the introduction, please shorten.
Response 7: Done as requested.
Comment 8: Line 151: ... up to 15 years ...
Response 8: The words "up to" were inserted as correctly advised.
Comment 9: Line 153: Despite it is well know that CTR is a safe technique with low complication rate you cannot state that from your study since you did not analysis the complication rate.
Response 9: Thank you, the reference to safety and low complication rate was omitted.

Round 2
Reviewer 2 Report
Comments and Suggestions for Authors
Dear authors,
thank you for the thorough review of your manuscript. Those are interesting results.
Author Response
We would like to express our sincere gratitude for your valuable comments and suggestions regarding our manuscript, “Long-Term Functional Outcomes and Quality of Life Following Carpal Tunnel Release Surgery.” We have carefully considered your feedback and have made the following revisions:
- Table Format and Font: All tables have been meticulously formatted to adhere to the guidelines of the MDPI journal. We have ensured consistency in font styles, sizes, and table structures throughout the manuscript.
- Limitations and Future Studies: We have incorporated a dedicated section within the Discussion to address the potential limitations arising from the non-participation of a portion of the study population.
- Sample Size Calculation: The text has been revised to accurately reflect that the final sample size of 186 participants was obtained and subsequently validated through a post-hoc power calculation. We have clarified that the initial sample size calculation was based on a conservative estimate and that the achieved sample size provided adequate statistical power for the study objectives.